# A Lower Bound for the Number of Linear Regions of Ternary ReLU Regression Neural Networks

**Yuta Nakahara**                                                    *y.nakahara@waseda.jp*
*Center for Data Science*
*Waseda University*

**Manabu Kobayashi**                                                 *mkoba@waseda.jp*
*Center for Data Science*
*Waseda University*

**Toshiyasu Matsushima**                                             *toshimat@waseda.jp*
*Department of Applied Mathematics*
*Waseda University*

**Reviewed on OpenReview:** *https://openreview.net/forum?id=Yg7tt1hWiF*

## Abstract

With the advancement of deep learning, reducing computational complexity and memory consumption has become a critical challenge, and ternary neural networks (NNs) that restrict parameters to $\{-1, 0, +1\}$ have attracted attention as a promising approach. While ternary NNs demonstrate excellent performance in practical applications such as image recognition and natural language processing, their theoretical understanding remains insufficient. In this paper, we theoretically analyze the expressivity of ternary NNs from the perspective of the number of linear regions. Specifically, we evaluate the number of linear regions of ternary regression NNs with Rectified Linear Unit (ReLU) for activation functions and prove that the number of linear regions increases polynomially with respect to network width and exponentially with respect to depth, similar to standard NNs. Moreover, we show that it suffices to first double the width, then either square the width or double the depth of ternary NNs with alternating ReLU and identity layers to achieve a lower bound on the maximum number of linear regions comparable to that of general ReLU regression NNs. When using ReLU in all the layers, a similar bound is obtained by further doubling the width. This provides a theoretical explanation, in some sense, for the practical success of ternary NNs.

## 1 Introduction

In recent years, with the rapid development of deep learning, neural networks (NNs) have achieved remarkable results in various fields. However, their large computational and memory consumption has become a serious barrier to applications in mobile devices and edge computing. Particularly, considering implementation in embedded systems that require real-time processing or in environments with limited computational resources, memory and computation reduction of NNs is an urgent issue.

As a promising approach to this problem, methods for discretizing NN parameters have been proposed. Specifically, methods that restrict network weights to binary $\{-1, +1\}$ or ternary $\{-1, 0, +1\}$ values, or quantize the output values of activation functions have been developed (Hubara et al., 2016; Liu et al., 2023). These methods have achieved performance comparable to conventional continuous-valued NNs in a wide range of tasks including image recognition (Rastegari et al., 2016; Liu et al., 2020), natural language processing (Bai et al., 2021; Wang et al., 2025), and speech recognition (Xiang et al., 2017), while successfully achieving significant reductions in computational complexity and memory usage. Particularly noteworthy

is the surprising fact that these discretized NNs can maintain high performance in practical tasks despite extremely restricting their parameters.

However, the theoretical understanding of why these discretization methods work effectively remains insufficient. The motivation of this study is to provide a theoretical explanation for the success of ternary NNs. When theoretically evaluating the performance of NNs, various perspectives can be considered, e.g, the expressivity, i.e., the complexity of functions representable by NNs, the empirical error for training data, and the generalization error when applying the trained model to new data. In this work, we evaluate the expressivity of ternary NNs, as restricting the parameter space raises concerns about its significant impact on the class of functions that NNs can represent. It should also be noted that, since expressivity is defined as a property of the NN itself independently of data, this study does not consider learning from data.

Various metrics can be considered for evaluating expressivity. For shallow NNs with three layers, Barron (1993) showed that, under certain conditions, any function can be universally approximated. On this basis, it was long believed that a depth of three layers is sufficient for NNs. Subsequently, as the superior performance of deep NNs was empirically demonstrated, theoretical researchers became interested in the advantages of increasing network depth. One of the early studies (Montúfar et al., 2014) on this topic evaluated the number of linear regions representable by NNs. This work showed that the maximum number of linear regions representable by deep NNs with ReLU activations grows polynomially in the width and exponentially in the depth of the network. Following this, various studies have evaluated different quantities related to linear regions (Pascanu et al., 2014; Serra et al., 2018; Hanin & Rolnick, 2019; Esaki et al., 2020). While these studies do not directly assess approximation accuracy of functions, it is intuitively clear that functions with an insufficient number of linear regions cannot approximate complex functions well. For instance, a function with only one linear region can not approximate a smooth curve well. Subsequently, following the approach of Barron, the approximation accuracy of functions within some classes is evaluated. For example, Yarotsky (2017) demonstrated that increasing the depth of a NN is more efficient than increasing its width for approximating functions in Sobolev spaces. Although Yarotsky (2017) did not directly used the results by Montúfar et al. (2014), his proof relies on a similar sawtooth (tent map) construction used by Montúfar et al. (2014) to count linear regions.

Thus, in the literature on the expressivity of deep NNs, the number of linear regions was historically evaluated first, then, approximation accuracy of functions was evaluated. In light of this background, we evaluate the number of linear regions of ternary NNs in this study. Our results may also provide some insights into evaluating the approximation accuracy of functions by ternary NNs, and it remains an important direction for future work. The main limitations of this study are as follows. While models such as BitNet b1.58 quantize not only the weights but also the outputs of activation functions, this aspect is outside the scope of this study.

The main contribution of this paper is to show that the maximum number of linear regions of ternary NNs also increases polynomially with respect to width and exponentially with respect to depth, similar to conventional NNs. More specifically, we prove that it suffices to first double the width, then either square the width or double the depth of ternary regression NNs with alternating ReLU and identity layers to obtain a lower bound on the maximum number of linear regions comparable to that for general ReLU regression NNs. When using ReLU in all the layers, a similar bound is obtained by further doubling the width. Although from the limited perspective of the number of linear regions of piecewise linear functions represented by ReLU NNs, and from the comparison between lower bounds on the maximum number of linear regions of conventional NNs and ternary NNs, these results provide one theoretical explanation for the practical success of ternary NNs.

The rest of this paper is structured as follows. Section 2 introduces notation for explaining this research and previous studies. Section 3 reviews existing research on the number of linear regions of general NNs. Section 4 states the main theorem regarding the number of linear regions of ternary NNs and provides its proof. Section 5 discusses the significance and limitations of the obtained results and concludes this research.

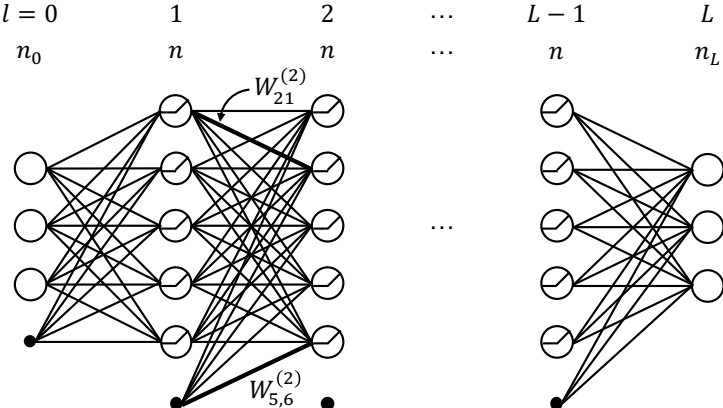

Figure 1: Illustration of NN. The weight of the edge extending from the $j$-th node of the $(l-1)$-th layer to the $i$-th node of the $l$-th layer corresponds to the $(i,j)$ component $W_{ij}^{(l)}$ of the weight matrix $\boldsymbol{W}^{(l)}$ of the linear function $\boldsymbol{W}^{(l)}\tilde{\boldsymbol{x}}$. The bent line inside the node represents that the activation function is ReLU. Each small black dot represents the constant term 1.

## 2    Preliminaries

**Definition 1** (Neural networks)**.** *Let an arbitrary natural number $L \in \mathbb{N}$ and natural numbers $n_l \in \mathbb{N}$ be given for integers $l = 0, 1, 2, \ldots, L$. Then, we call the following function $\boldsymbol{F_\theta} : \mathbb{R}^{n_0} \to \mathbb{R}^{n_L}$, which is expressed as a composition of linear functions[1] $\boldsymbol{f}^{(l)} : \mathbb{R}^{n_{l-1}} \to \mathbb{R}^{n_l}; \boldsymbol{x} \mapsto \boldsymbol{W}^{(l)}\tilde{\boldsymbol{x}}$, where $\tilde{\boldsymbol{x}}$ represents the vector $[x_1, x_2, \ldots, x_{n_{l-1}}, 1]^\top$, and nonlinear functions $\boldsymbol{g}^{(l)} : \mathbb{R}^{n_l} \to \mathbb{R}^{n_l}$, a NN of depth $L$ with width $n_l$ at the $l$-th layer:*

$$\boldsymbol{F_\theta}(\boldsymbol{x}) = \boldsymbol{f}^{(L)} \circ \boldsymbol{g}^{(L-1)} \circ \boldsymbol{f}^{(L-1)} \circ \cdots \circ \boldsymbol{g}^{(1)} \circ \boldsymbol{f}^{(1)}(\boldsymbol{x}). \tag{1}$$

*Here, $n_0$ and $n_L$ represent the dimensions of input and output, respectively, and $\boldsymbol{\theta}$ represents all parameters $\{\boldsymbol{W}^{(l)}\}_{l=1}^{L}$ of the NN. In this paper, we only deal with regression NNs where the final layer is a linear function. The nonlinear function $\boldsymbol{g}^{(l)}$ is called an activation function. In particular, we call regression NNs that use the following ReLU function for all the activation functions ReLU Regression NNs in this paper.*

$$g_j^{(l)}(\boldsymbol{x}) = \max\{0, x_j\}, \tag{2}$$

*where $g_j^{(l)}(\boldsymbol{x})$ and $x_j$ represent the $j$-th components of $\boldsymbol{g}^{(l)}(\boldsymbol{x})$ and $\boldsymbol{x}$, respectively. Also, we call NNs where all the parameters of the linear function $\boldsymbol{f}^{(l)}$ take only values from $\{1, 0, -1\}$ ternary NNs in this paper.*

Limiting NN weights to ternary values is also performed in BitNet b1.58 (Wang et al., 2025). While Bit-Net b1.58 additionally quantizes the output of activation functions, as mentioned in the introduction, this paper does not deal with quantization of activation functions. Evaluating the effect of activation function quantization on NN expressivity is a future research topic.

NNs can be represented as graphs as shown in Fig. 1. Each small black dot represents the constant term 1. In this paper, we do not distinguish the coefficients for input variables and those for constant terms, i.e., biases. Both are represented by $W_{i,j}^{(l)}$ and called *weights*. In other words, if NNs are ternary, biases are also restricted to $\{1, 0, -1\}$ in our setting. If an activation function used at a node is ReLU, we represent it by drawing the bent line inside the node.

**Definition 2** (Linear regions (Montúfar et al., 2014))**.** *For a function $\boldsymbol{f} : D \to \mathbb{R}^n$, we say that $U \subset D$ is a linear region of $\boldsymbol{f}$ if the following holds:*

---

[1]In this paper, linear functions refer to functions that form some hyperplane and do not necessarily pass through the origin. More precisely, such functions are called affine functions, but following the notation of previous research (Montúfar et al., 2014), we call them linear functions in this paper.

- *The restriction[2] of $\boldsymbol{f}$ to $U$, $\boldsymbol{f}|_U : U \to \mathbb{R}^n$, is a linear function.*

- *For any $V \subset D$ satisfying $V \supsetneq U$, $\boldsymbol{f}|_V$ is not a linear function.*

*That is, a locally maximal subset $U$ of $D$ where $\boldsymbol{f}$ becomes a linear function within that region is called a linear region of $\boldsymbol{f}$. Also, we denote the set of all linear regions of $\boldsymbol{f}$ as $\mathcal{L}(\boldsymbol{f})$ in this paper.*

**Example 1** (Linear regions of absolute value function)**.** *The linear regions of the absolute value function $f(x) = |x|$ are $(-\infty, 0]$ and $[0, \infty)$, so $\mathcal{L}(f) = \{(-\infty, 0], [0, \infty)\}$. Note that linear regions are closed sets, by its definition.*

Here, note that when $\boldsymbol{f} : D \to \mathbb{R}^n$ is a piecewise linear function, the following holds:

$$\bigcup_{U \in \mathcal{L}(\boldsymbol{f})} U = D. \tag{3}$$

## 3 Previous Studies

Conventionally, it is known that the following holds for the number of linear regions $|\mathcal{L}(\boldsymbol{F})|$ of a ReLU Regression NN $\boldsymbol{F} : \mathbb{R}^{n_0} \to \mathbb{R}^{n_L}$ with depth $L$ and width $n$ at each layer.

**Proposition 1** (A lower bound for the number of linear regions of ReLU Regression NNs (Montúfar et al., 2014))**.** *For a ReLU Regression NN $\boldsymbol{F_\theta} : \mathbb{R}^{n_0} \to \mathbb{R}^{n_L}$ with depth $L$ and width $n$ at each layer, let $p = \lfloor \frac{n}{n_0} \rfloor$. Then the following holds:*

$$\max_{\boldsymbol{\theta}} |\mathcal{L}(\boldsymbol{F_\theta})| \geq p^{n_0(L-1)}. \tag{4}$$

This suggests that the number of linear regions of NNs increases in polynomial order with respect to network width and in exponential order with respect to depth, and is considered one piece of evidence showing that deepening the depth is more effective than widening the width for enhancing the expressivity of NNs.

The proof of Proposition 1 in (Montúfar et al., 2014) is given by specifically constructing a ReLU Regression NN such that the number of linear regions becomes $p^{n_0(L-1)}$. However, while Montúfar et al. (2014) shows the construction procedure of the ReLU Regression NN, it does not concisely state the mathematical formulas of the linear functions of each layer constructed by this procedure. To make it easier to use in the proof of our theorem described later, we express this using mathematical formulas here. In the ReLU Regression NN constructed by the method of (Montúfar et al., 2014), the $n$ nodes of each intermediate layer are divided into $p$ groups of $n_0$ nodes each, and the linear function $f^{(l)}_{(i-1)n_0+j}(\boldsymbol{x})$ corresponding to the $j$-th node of the $i$-th group in the $l$-th layer is expressed by the following formula (for the remaining $n - pn_0$ nodes, all weights are set to 0, so that 0 is identically output).

- **The first layer:** for $l = 1$, $i = 1, 2, \ldots, p$ and $j = 1, 2, \ldots, n_0$, the $((i-1)n_0 + j)$-th component $f^{(1)}_{(i-1)n_0+j}(\boldsymbol{x})$ of $\boldsymbol{f}^{(1)}(\boldsymbol{x})$ is expressed as follows:

$$f^{(1)}_{(i-1)n_0+j}(\boldsymbol{x}) = \begin{cases} px_j, & i = 1, \\ 2px_j - 2(i-1), & 2 \leq i \leq p. \end{cases} \tag{5}$$

- **Intermediate layers:** for $l = 2, 3, \ldots, L-1$, $i = 1, 2, \ldots, p$ and $j = 1, 2, \ldots, n_0$, the $((i-1)n_0+j)$-th component $f^{(l)}_{(i-1)n_0+j}(\boldsymbol{x})$ of $\boldsymbol{f}^{(l)}(\boldsymbol{x})$ is expressed as follows:

$$f^{(l)}_{(i-1)n_0+j}(\boldsymbol{x}) = \begin{cases} p\sum_{k=1}^{p}(-1)^{k-1}x_{(k-1)n_0+j}, & i = 1, \\ 2p\sum_{k=1}^{p}(-1)^{k-1}x_{(k-1)n_0+j} - 2(i-1), & 2 \leq i \leq p. \end{cases} \tag{6}$$

---

[2]A function that has the same input-output mapping as $\boldsymbol{f}$ but is defined only on $U$.

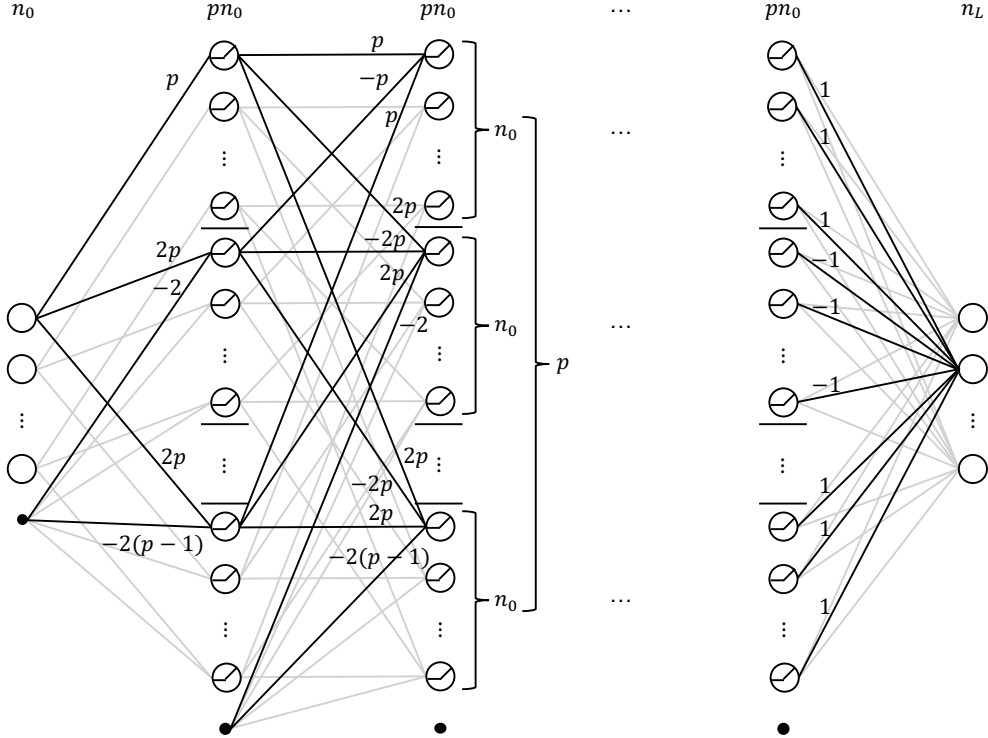

Figure 2: ReLU Regression NN with number of linear regions equal to $p^{n_0(L-1)}$.

- **The last layer:** for $l = L$, $m = 1, 2, \ldots, n_L$, the $m$-th component $f_m^{(L)}(\boldsymbol{x})$ of $\boldsymbol{f}^{(L)}(\boldsymbol{x})$ is expressed as follows:

$$f_m^{(L)}(\boldsymbol{x}) = \sum_{j=1}^{n_0} \sum_{k=1}^{p} (-1)^{k-1} x_{(k-1)n_0+j}. \tag{7}$$

We can represent these as a graph shown in Fig. 2.

Although it is merely a rewrite of the proof in (Montúfar et al., 2014) using equations equation 5, equation 6, and equation 7, the proof that the number of linear regions of this ReLU Regression NN is indeed $p^{n_0(L-1)}$ is summarized in Appendix.

## 4    Main Results

Before explaining the main results, we show that bounded integer weight regression NNs can be represented by ternary NNs. First, any edge of a bounded integer weight NN can be represented by a ternary regression NN with identity activation functions. Consider a NN where there exists some natural number $M$ such that the weights of linear functions are represented by integers in the range from $-M$ to $M$. Suppose one edge has integer weight $w$ as shown in Fig. 3 (a). That is, suppose this NN partially includes the process of multiplying the input value by $w$. This process of multiplying the input value by $w$ can be represented in a two-layer ternary regression NN with identity activation functions, where the number of nodes in each layer

Figure 3: (a): A certain edge of a bounded integer weight NN. Here, the maximum weight is $M = 5$ and the weight in this example is $w = 3$. (b-1): Representation of (a) by a ternary regression NN with identity activation functions. (b-2): Abbreviated notation of (b-1). Note that the number near by the triple lines between the square node and the round node represents not an edge weight but the sum of the original edge weights in (b-1).

being $n_0 = 1$, $n_1 = M$, $n_2 = 1$, by setting

$$f_j^{(1)}(x) = x, \quad (j = 1, 2, \ldots, M) \tag{8}$$

$$\boldsymbol{g}^{(1)}(\boldsymbol{y}) = \boldsymbol{y}, \tag{9}$$

$$f^{(2)}(\boldsymbol{z}) = \sum_{j=1}^{|w|} \text{sign}(w) z_j \tag{10}$$

where $\text{sign}(w)$ represents the sign of $w$. This function is also represented as a graph shown in Fig. 3 (b-1). Further, as shown in Fig. 3 (b-2), we represent the $M$ nodes in the intermediate layer in this graph by one square node and the $M$ edges to or from these nodes by triple lines. Note that the number near by the triple lines represents not an edge weight but the sum of the original edge weights in Fig. 3 (b-1).

Furthermore, when representing not an edge but a bounded integer weight NN with a ternary regression NN with identity activation functions, it is sometimes possible to represent it with fewer nodes than replacing each edge with the aforementioned transformation by combining the nodes of the ternary NN. For example, Fig. 4 shows ternary regression NNs with identity activation functions equivalent to a bounded integer weight NN (a). In Fig. 4, (b) shows the ternary regression NN with identity activation functions by trivial transformation, (c-1) shows another representation of (a) by transformation with fewer nodes, and (c-2) shows its abbreviated notation. In Fig. 4 (c-2), edges extend from a square node to multiple nodes, with some numbers attached. Each of these edges represents the $M$ edges extending from the $M$ nodes in (c-1), which correspond to the square node in (c-2), and the number represents the sum of the weights of those $M$ edges.

Next, we state the main result of this research and its proof. In the following, for a ReLU Regression NN with depth $L$ and width $n$, we consider a ternary regression NN with the same depth $L$ and width $n$, and evaluate the lower bound of the maximum number of linear regions that can be represented by adjusting the edge weights. The proof strategy is as follows. We utilize the fact that the coefficients and biases of the linear functions equation 5, equation 6, equation 7 used in the proof of Proposition 1 are bounded integers. We also leverage the relationship between bounded integer weight NNs and ternary regression NNs with identity activation functions mentioned above. Based on these observations, we first construct a ternary regression NN with alternating ReLU and identity layers that represents functions of the same form as equation 5, equation 6, equation 7. The lower bound when using ReLU in all the layers are discussed later. The following is the main result of this research.

**Theorem 1** (A lower bound of the number of linear regions of ternary NN with ReLU after even layer). *For a ternary NN $\boldsymbol{F_\theta} : \mathbb{R}^{n_0} \to \mathbb{R}^{n_L}$ with depth $L$ and width $n$ at each layer, where the activation function of odd-numbered layers is the identity function and the activation function of even-numbered layers is ReLU, let $L'$ be the maximum odd number less than or equal to $L$ and $q = \lfloor \frac{n}{2(n_0+1)} \rfloor$. Then the following holds:*

$$\max_{\boldsymbol{\theta}} |\mathcal{L}(\boldsymbol{F_\theta})| \geq q^{\frac{1}{2} n_0 (L'-1)}. \tag{11}$$

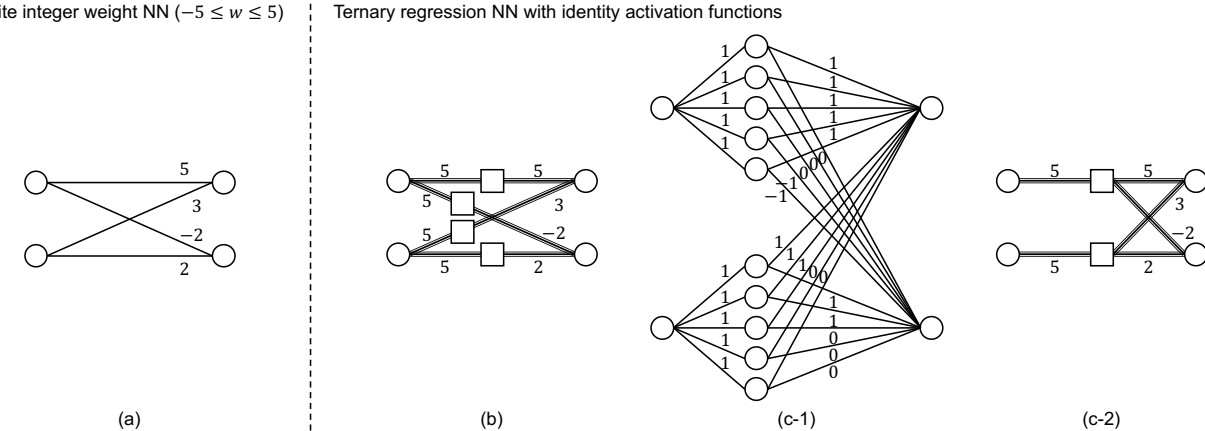

Figure 4: (a): A bounded integer weight NN where weight $w$ satisfies $-5 \leq w \leq 5$. (b): Representation of (a) by a ternary regression NN with identity activation functions through trivial transformation. (c-1): Representation of (a) by a ternary regression NN with identity activation functions through transformation with fewer nodes. (c-2): Abbreviated notation of (c-1). Note that the number near by the triple lines between the square node and the round node represents not an edge weight but the sum of the original edge weights.

*Proof.* By constructing the ternary regression NN with alternating ReLU and identity layers shown in Fig. 5, the function represented by this NN becomes the function obtained by replacing $p$ with $q = \lfloor \frac{n}{2(n_0+1)} \rfloor$ in equation 5, equation 6, equation 7. In the following, we describe the specific construction method of this NN and the mathematical expression of the function represented by it.

Let $L'$ be the maximum odd number less than or equal to $L$. If $L$ is even, we set the final layer to be an identity transformation, and hereafter we construct a ternary regression NN with depth $L'$ and width $n$ at each layer, where ReLU and identity layers are alternately used. Setting $q = \lfloor \frac{n}{2(n_0+1)} \rfloor$, we divide the $n$ nodes of the odd layers of this NN into $n_0 + 1$ groups of $2q$ nodes each. In Fig. 5, these $2q$ nodes are represented by one square node. For the remaining $n - 2q(n_0 + 1)$ nodes, all weights are set to 0, making them functions that identically output 0. For the nodes of even layers, we create $q$ groups of $n_0$ nodes each, and for the remaining $n - qn_0$ nodes, all weights are set to 0, so that they identically output 0. First, since equation equation 7 already takes only values from $\{1, 0, -1\}$ for any weights, we can use the linear function obtained by replacing $p$ with $q$ in equation equation 7 for the $L'$-th layer. Next, for $l = 1, 2, \ldots, \frac{1}{2}(L'-1)$, we define the $(2l-1)$-th layer and the $2l$-th layer as follows:

- For $l = 1$:
  - Definition of $\boldsymbol{f}^{(1)}$: For $j = 1, 2, \ldots, n_0 + 1$ and $k = 1, 2, \ldots, 2q$, define the $(2q(j-1) + k)$-th component $f^{(1)}_{2q(j-1)+k}(\boldsymbol{x})$ of $\boldsymbol{f}^{(1)}(\boldsymbol{x})$ as follows:

$$f^{(1)}_{2q(j-1)+k}(\boldsymbol{x}) = \begin{cases} x_j, & j = 1, 2, \ldots, n_0 \\ 1, & j = n_0 + 1. \end{cases} \tag{12}$$

  - Definition of $\boldsymbol{f}^{(2)}$: For $i = 1, 2, \ldots, q$ and $j = 1, 2, \ldots, n_0$, define the $((i-1)n_0 + j)$-th component $f^{(2)}_{(i-1)n_0+j}(\boldsymbol{x})$ of $\boldsymbol{f}^{(2)}(\boldsymbol{x})$ as follows:

$$f^{(2)}_{(i-1)n_0+j}(\boldsymbol{x}) = \begin{cases} \sum_{k=1}^{q} x_{2q(j-1)+k}, & i = 1 \\ \sum_{k=1}^{2q} x_{2q(j-1)+k} + \sum_{k=1}^{2(i-1)} (-1), & i = 2, 3, \ldots, q. \end{cases} \tag{13}$$

- For $l = 2, 3, \ldots, \frac{1}{2}(L'-1)$:

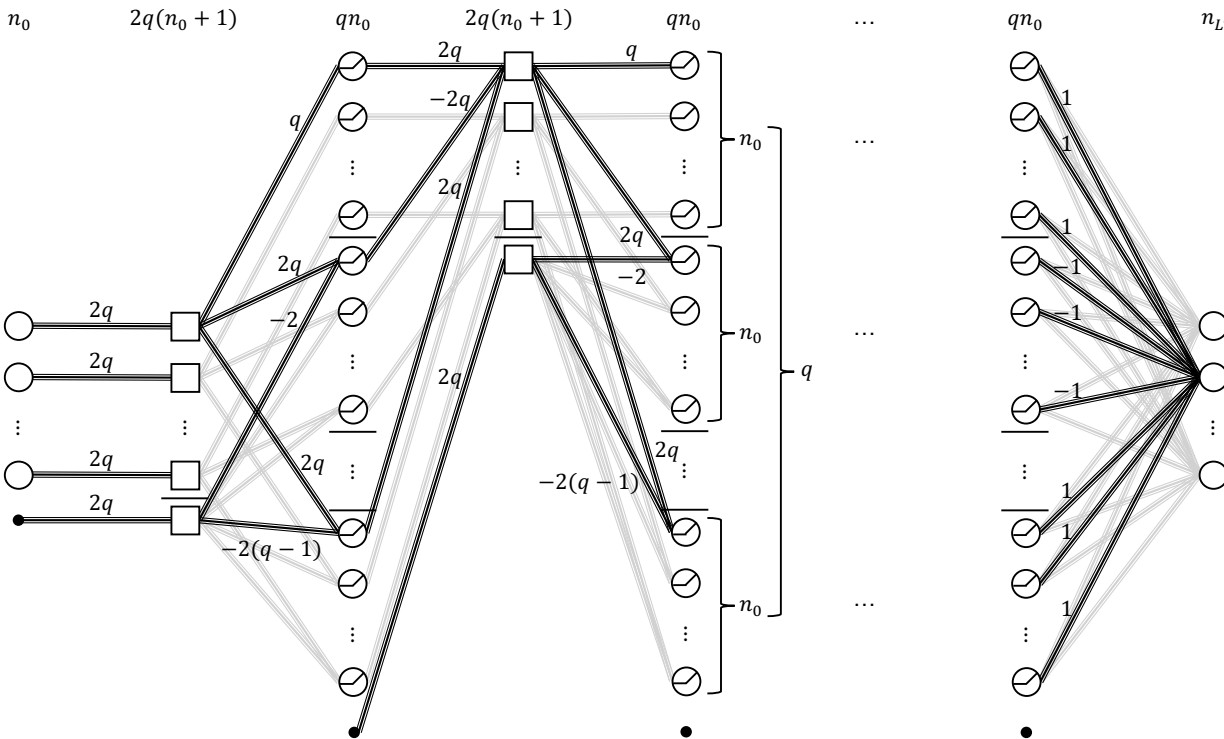

Figure 5: A ternary regression NN whose number of linear regions equal to $q^{\frac{1}{2}n_0(L'-1)}$, where ReLU and identity layers are alternately used. Note that the number near by the triple lines between the square node and the round node represents not an edge weight but the sum of the original edge weights.

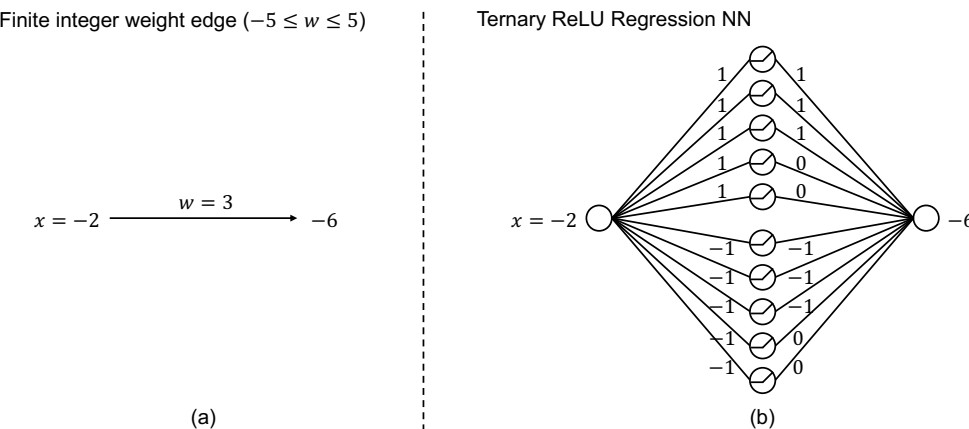

Figure 6: (a): A certain edge of a bounded integer weight NN (the same as in Fig. 3 (a)). Here, the maximum weight is $M = 5$ and the weight in this example is $w = 3$. (b): Representation of (a) by a ternary ReLU Regression NN

- Definition of $\boldsymbol{f}^{(2l-1)}$: For $j = 1, 2, \ldots, n_0 + 1$ and $k = 1, 2, \ldots, 2q$, define the $(2q(j-1)+k)$-th component $f_{2q(j-1)+k}^{(2l-1)}(\boldsymbol{x})$ of $\boldsymbol{f}^{(2l-1)}(\boldsymbol{x})$ as follows:

$$f_{2q(j-1)+k}^{(2l-1)}(\boldsymbol{x}) = \begin{cases} \sum_{m=1}^{q} (-1)^{m-1} x_{(m-1)n_0+j}, & j = 1, 2, \ldots, n_0 \\ 1, & j = n_0 + 1. \end{cases} \tag{14}$$

- Definition of $\boldsymbol{f}^{(2l)}$: For $i = 1, 2, \ldots, q$ and $j = 1, 2, \ldots, n_0$, define the $((i-1)n_0+j)$-th component $f_{(i-1)n_0+j}^{(2l)}(\boldsymbol{x})$ of $\boldsymbol{f}^{(2l)}(\boldsymbol{x})$ as follows:

$$f_{(i-1)n_0+j}^{(2l)}(\boldsymbol{x}) = \begin{cases} \sum_{k=1}^{q} x_{2q(j-1)+k}, & i = 1 \\ \sum_{k=1}^{2q} x_{2q(j-1)+k} + \sum_{k=1}^{2(i-1)} (-1), & i = 2, 3, \ldots, q. \end{cases} \tag{15}$$

For activation functions, we use the identity function for odd layers and ReLU for even layers.

Since the activation function of odd layers is the identity function, when we substitute $\boldsymbol{f}^{(1)}$ into $\boldsymbol{f}^{(2)}$, we can see that $\boldsymbol{f}^{(2)} \circ \boldsymbol{f}^{(1)}$ of the ternary regression NN equals the function obtained by replacing $p$ with $q$ in the linear function equation 5 of the first layer of a regular ReLU Regression NN. Also, for $l = 2, 3, \ldots, \frac{1}{2}(L'-1)$, when we substitute $\boldsymbol{f}^{(2l-1)}$ into $\boldsymbol{f}^{(2l)}$, we can see that $\boldsymbol{f}^{(2l)} \circ \boldsymbol{f}^{(2l-1)}$ of the ternary regression NN equals the function obtained by replacing $p$ with $q$ in the linear function equation 6 of the $l$-th layer of a regular NN. Therefore, the ternary regression NN shown here achieves the lower bound of the number of linear regions obtained by replacing $p$ with $q$ and $L - 1$ with $\frac{1}{2}(L'-1)$ in the right-hand side of equation equation 4 for a regular ReLU Regression NN. That is, the number of linear regions of this ternary regression NN becomes $q^{\frac{1}{2}n_0(L'-1)}$. $\qquad\square$

**Remark 1.** *Comparing equation equation 4 and equation equation 11, roughly speaking, to obtain a lower bound of the maximum number of linear regions comparable to that for general ReLU Regression NNs, it suffices to first double the width, then square the width or double the depth of ternary NNs where the activation function of odd-numbered layers is the identity function and the activation function of even-numbered layers is ReLU.*

**Remark 2.** *The identity function can be represented by two ReLU functions as follows: $x = \text{ReLU}(x) - \text{ReLU}(-x)$. Using this equation, Theorem 1 can be extended to ternary regression NNs with ReLU in all the layers. For example, a finite integer weight edge as shown in Fig. 6 (a), which is the same as in Fig 3 (a), can be represented by a ternary ReLU Regression NN as shown in Fig. 6 (b). By replacing all the square nodes in Fig. 5 with this method, we can achieve a bound similar to Theorem 1 using a ternary ReLU Regression NN with double the width of the ternary NN with alternating ReLU and identity layers.*

## 5 Conclusion and Future Work

We theoretically evaluated the expressivity of ternary NNs, which have achieved great practical success in memory and computation reduction of NNs, from the perspective of the number of linear regions. As a result, it was shown that the expressivity of ternary ReLU Regression NNs increases polynomially with respect to network width and exponentially with respect to depth, similar to general ReLU Regression NNs. Furthermore, it was found that it suffices to first double the width, then square the width or double the depth of ternary regression NNs with alternating ReLU and identity layers to obtain a lower bound on the maximum number of linear regions comparable to that for general ReLU Regression NNs. When using ReLU in all the layers, a similar bound is obtained by further doubling the width. We believe this provides an explanation of a part of the reason for the practical success of ternary NNs, albeit from the limited perspective of the number of linear regions of piecewise linear functions represented by ReLU NNs.

However, in actual applications such as BitNet b1.58, the output of activation functions are also quantized for further memory and computation reduction. Our method cannot be directly applied to such NNs. Theoretical evaluation of the expressivity of such NNs is a future research topic. Moreover, the evaluation of approximation accuracy of functions by ternary NNs remains an important direction for future work. Applying the expressivity bound derived in this paper to a practical example is also important, e.g., investigating how well a real-world ternary network satisfies the bound.

### Acknowledgments

We thank Professor Suko at Waseda University for providing the motivation for this research. This work was supported in part by JSPS KAKENHI Grant Numbers JP22K02811, JP23K03863, JP23K04293, JP24H00370 and 26K17386.

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

# A  Proof of Proposition 1

Before proving Proposition 1, we show the following Lemma.

**Lemma 1** (The number of linear regions of a composite function of piecewise linear functions). *Let $\boldsymbol{g} : D_1 \to D_2$ be a piecewise linear function, and suppose that for any linear region $U \in \mathcal{L}(\boldsymbol{g})$ of $\boldsymbol{g}$, $\boldsymbol{g}|_U$ is a bijection from $U$ to $D_2$. Let $\boldsymbol{f} : D_2 \to \mathbb{R}^n$ be a piecewise linear function. Note that the domain of $\boldsymbol{f}$ coincides with the range of $\boldsymbol{g}$. Then, the following holds for the number of linear regions of the composite function $\boldsymbol{f} \circ \boldsymbol{g}$:*

$$|\mathcal{L}(\boldsymbol{f} \circ \boldsymbol{g})| = |\mathcal{L}(\boldsymbol{f})||\mathcal{L}(\boldsymbol{g})|. \tag{16}$$

*Proof of Lemma 1.* For any $U \in \mathcal{L}(\boldsymbol{g})$ and any $V \in \mathcal{L}(\boldsymbol{f})$, consider the inverse image of $V$ under $\boldsymbol{g}|_U$, denoted $\boldsymbol{g}|_U^{-1}(V)$. On $\boldsymbol{g}|_U^{-1}(V)$, $\boldsymbol{f} \circ \boldsymbol{g}|_U$ is clearly a linear function. Also, on the set $\boldsymbol{g}|_U^{-1}(V) \cup \{x\}$ obtained by adding any point $x \in U \setminus \boldsymbol{g}|_U^{-1}(V)$ to $\boldsymbol{g}|_U^{-1}(V)$, $\boldsymbol{f} \circ \boldsymbol{g}|_U$ does not become a linear function. This is because, due to the bijectivity of $\boldsymbol{g}|_U$, we have $\boldsymbol{g}|_U(\boldsymbol{g}|_U^{-1}(V) \cup \{x\}) \supsetneq V$, and by the definition of linear region $V$, $\boldsymbol{f}|_{\boldsymbol{g}|_U(\boldsymbol{g}|_U^{-1}(V) \cup \{x\})}$ does not become a linear function. Therefore, $\boldsymbol{g}|_U^{-1}(V)$ is a linear region of $\boldsymbol{f} \circ \boldsymbol{g}|_U$.

Also, the following holds:

$$U = \boldsymbol{g}|_U^{-1}(D_2) \qquad \because \boldsymbol{g}|_U \text{ is a bijection from } U \text{ to } D_2 \qquad (17)$$

$$= \boldsymbol{g}|_U^{-1}\left(\bigcup_{V \in \mathcal{L}(\boldsymbol{f})} V\right) \qquad \because \boldsymbol{f} \text{ is a piecewise linear function} \qquad (18)$$

$$= \bigcup_{V \in \mathcal{L}(\boldsymbol{f})} \boldsymbol{g}|_U^{-1}(V). \qquad \because \boldsymbol{g}|_U \text{ is a bijection from } U \text{ to } D_2 \qquad (19)$$

This indicates that $U$ is divided into $|\mathcal{L}(\boldsymbol{f})|$ linear regions $\boldsymbol{g}|_U^{-1}(V)$ of $\boldsymbol{f} \circ \boldsymbol{g}|_U$. Since the same holds for any $U \in \mathcal{L}(\boldsymbol{g})$, we have $|\mathcal{L}(\boldsymbol{f} \circ \boldsymbol{g})| = |\mathcal{L}(\boldsymbol{f})||\mathcal{L}(\boldsymbol{g})|$. $\qquad\qquad\square$

Next, we prove Proposition 1.

*Proof of Proposition 1.* First, we define the following three functions $\tilde{\boldsymbol{f}} : [0,1]^{n_0} \to \mathbb{R}^{pn_0}$, $\tilde{\tilde{\boldsymbol{f}}} : \mathbb{R}^{pn_0} \to [0,1]^{n_0}$, and $\tilde{\tilde{\tilde{\boldsymbol{f}}}} : [0,1]^{n_0} \to \mathbb{R}^{n_L}$. Note the domain and range of each function.

- Definition of $\tilde{\boldsymbol{f}} : [0,1]^{n_0} \to \mathbb{R}^{pn_0}$: For any $i = 1, 2, \ldots, p$ and $j = 1, 2, \ldots, n_0$, define the $((i-1)n_0+j)$-th component $\tilde{f}_{(i-1)n_0+j}(\boldsymbol{x})$ of $\tilde{\boldsymbol{f}}(\boldsymbol{x})$ as follows:

$$\tilde{f}_{(i-1)n_0+j}(\boldsymbol{x}) = \begin{cases} px_j, & i = 1 \\ 2px_j - 2(i-1), & i = 2, 3, \ldots, p. \end{cases} \qquad (20)$$

- Definition of $\tilde{\tilde{\boldsymbol{f}}} : \mathbb{R}^{pn_0} \to [0,1]^{n_0}$: For any $j = 1, 2, \ldots, n_0$, define the $j$-th component $\tilde{\tilde{f}}_j(\boldsymbol{x})$ of $\tilde{\tilde{\boldsymbol{f}}}(\boldsymbol{x})$ as follows:

$$\tilde{\tilde{f}}_j(\boldsymbol{x}) = \sum_{k=1}^{p} (-1)^{k-1} x_{(k-1)n_0+j}. \qquad (21)$$

- Definition of $\tilde{\tilde{\tilde{\boldsymbol{f}}}} : [0,1]^{n_0} \to \mathbb{R}^{n_L}$: For any $m = 1, 2, \ldots, n_L$, define the $m$-th component $\tilde{\tilde{\tilde{f}}}_m(\boldsymbol{x})$ of $\tilde{\tilde{\tilde{\boldsymbol{f}}}}(\boldsymbol{x})$ as follows:

$$\tilde{\tilde{\tilde{f}}}_m(\boldsymbol{x}) = \sum_{j=1}^{n_0} x_j. \qquad (22)$$

For simplicity, we restrict the domain of the NN to $[0,1]^{n_0}$. Then, $\boldsymbol{F_\theta}(\boldsymbol{x})$ can be expressed using these functions and ReLU $\boldsymbol{g}$ as follows:

$$\boldsymbol{F_\theta}(\boldsymbol{x}) = \boldsymbol{f}^{(L)} \circ \boldsymbol{g}^{(L-1)} \circ \boldsymbol{f}^{(L-1)} \circ \cdots \circ \boldsymbol{g}^{(1)} \circ \boldsymbol{f}^{(1)}(\boldsymbol{x}) \qquad (23)$$

$$= \tilde{\tilde{\tilde{\boldsymbol{f}}}} \circ \underbrace{\tilde{\tilde{\boldsymbol{f}}} \circ \boldsymbol{g} \circ \tilde{\boldsymbol{f}}}_{\boldsymbol{h}} \circ \underbrace{\tilde{\tilde{\boldsymbol{f}}} \circ \boldsymbol{g} \circ \tilde{\boldsymbol{f}}}_{\boldsymbol{h}} \circ \cdots \circ \underbrace{\tilde{\tilde{\boldsymbol{f}}} \circ \boldsymbol{g} \circ \tilde{\boldsymbol{f}}}_{\boldsymbol{h}} \qquad (24)$$

$$= \tilde{\tilde{\tilde{\boldsymbol{f}}}} \circ \boldsymbol{h} \circ \boldsymbol{h} \circ \cdots \circ \boldsymbol{h}. \qquad (25)$$

Note that $\boldsymbol{h} = \tilde{\tilde{\boldsymbol{f}}} \circ \boldsymbol{g} \circ \tilde{\boldsymbol{f}}$ is a function from $[0,1]^{n_0}$ to $[0,1]^{n_0}$. Also, the $j$-th component of $\boldsymbol{h}(\boldsymbol{x})$ is expressed as

$$h_j(\boldsymbol{x}) = \max\{0, px_j\} - \max\{0, 2px_j - 2\} + \\ \cdots + (-1)^{p-1}\max\{0, 2px_j - 2(p-1)\} \qquad (26)$$

and is a one-dimensional piecewise linear function that depends only on $x_j$, as shown in Fig. 7.

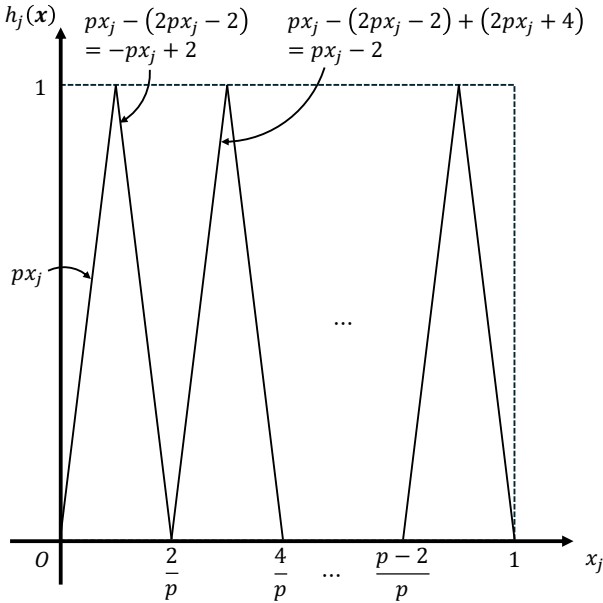

Figure 7: A graph of $h_j(\boldsymbol{x})$

Therefore, for any $t \in \{0, 1, \ldots, p-1\}$, an interval $\left[\frac{t}{p}, \frac{t+1}{p}\right]$ becomes a linear region of $h_j$, and $h_j|_{\left[\frac{t}{p}, \frac{t+1}{p}\right]}$ becomes a bijection to $[0, 1]$. Since $\boldsymbol{h}$ is a function consisting of $h_j$ as each component, for any $(t_1, t_2, \ldots, t_{n_0}) \in \{0, 1, \ldots, p-1\}^{n_0}$, the Cartesian product $\prod_{j=1}^{n_0} \left[\frac{t_j}{p}, \frac{t_j+1}{p}\right]$ becomes a linear region of $\boldsymbol{h}$, and $\boldsymbol{h}|_{\prod_{j=1}^{n_0} \left[\frac{t_j}{p}, \frac{t_j+1}{p}\right]}$ becomes a bijection to $[0, 1]^{n_0}$.

Here, since $|\mathcal{L}(\tilde{\tilde{\boldsymbol{f}}})| = 1$ and $|\mathcal{L}(\boldsymbol{h})| = |\{0, 1, \ldots, p-1\}^{n_0}| = p^{n_0}$, using Lemma 1, the following holds:

$$|\mathcal{L}(\tilde{\tilde{\boldsymbol{f}}} \circ \boldsymbol{h})| = |\mathcal{L}(\tilde{\tilde{\boldsymbol{f}}})||\mathcal{L}(\boldsymbol{h})| = 1 \cdot p^{n_0} = p^{n_0}. \tag{27}$$

Similarly, the following also holds:

$$|\mathcal{L}(\tilde{\tilde{\boldsymbol{f}}} \circ \boldsymbol{h} \circ \boldsymbol{h})| = |\mathcal{L}(\tilde{\tilde{\boldsymbol{f}}} \circ \boldsymbol{h})||\mathcal{L}(\boldsymbol{h})| = p^{n_0} \cdot p^{n_0} = p^{2n_0}. \tag{28}$$

By repeating this recursively, we obtain

$$|\mathcal{L}(\boldsymbol{F_\theta})| = |\mathcal{L}(\tilde{\tilde{\boldsymbol{f}}} \circ \boldsymbol{h} \circ \boldsymbol{h} \circ \cdots \circ \boldsymbol{h})| = p^{n_0(L-1)} \tag{29}$$

Therefore, Proposition 1 is proven. □

