# OpenReview forum: "A Lower Bound for the Number of Linear Regions of Ternary ReLU Regression Neural Networks"
_TMLR — Accepted by TMLR_

### Review · Reviewer_nWkw · 2026-01-27

**Summary Of Contributions:**

The manuscript presents a single mathematical result on the number of linear regions attainable in the output space of the ReLU neural network with weights constrained to three values: -1, 0, or 1, in terms of the network's depth and width. This result is compared with a seminal one of Montufar et al. (2014) on the number of linear regions of a ReLU network with (standard) real-valued weights

**Additional Comments:**

I think the following paper deserves to be cited, as it provides a similar analysis of ReLU networks to that of Montufar et al. (2014), but in terms of approximation error instead of the number of regions:
Dmitry Yarotsky (2017). Error bounds for approximations with deep ReLU networks. Neural Networks.

**Audience:**

Yes

**Audience Explanation:**

To my knowledge, the result is new. Given the empirical benefit of using a quantized network (in terms of memory reduction), this result is of interest.

**Broader Impact Concerns:**

-

**Claims And Evidence:**

Yes

**Claims Explanation:**

The paper contribution is modest but interesting. It provides exactly the result its title announces. The main ideas are well explained, and the proofs are detailed. I appreciate that the authors do not oversell their results. It

**Requested Changes:**

### Critical (but easy to fix)

Please carefully proofread the bibliography. When a paper has been accepted after a peer-reviewed process, it is a duty to cite the published version instead of the arXiv preprint:
- Courbariaux et al. (2016) corresponds to this NeurIPS paper: https://proceedings.neurips.cc/paper/2016/hash/d8330f857a17c53d217014ee776bfd50-Abstract.html
- Li et al. (2022) is an ICASSP 2023 paper.
- Pascanu et al. (2014) is an ICLR paper

### Further recommendations

1. I think the following paper deserves to be cited, as it provides a similar analysis of ReLU networks to that of Montufar et al. (2014), but in terms of approximation error instead of the number of regions:
> Dmitry Yarotsky (2017). Error bounds for approximations with deep ReLU networks. Neural Networks.

2. The depiction of the neural networks' decompositions in the figures is really helpful, but it took me a bit of time to grasp the meaning of the edge labels, as the meanings differ across figures. I would suggest clarifying the figures' explanation.

---

> ### Author Response · Authors · 2026-03-12
>
> We sincerely thank the reviewer for their thorough and constructive feedback. We are pleased that the reviewer finds our contribution interesting and acknowledges the novelty and clarity of our work. Below, we address each point raised.
>
> ### Critical Changes (Bibliography Corrections)
>
> We apologize for the oversight in our bibliography and have corrected all citations to reflect their published versions:
>
> * Courbariaux et al. (2016): Updated to cite the [NeurIPS 2016 proceedings version](https://proceedings.neurips.cc/paper_files/paper/2016/file/d8330f857a17c53d217014ee776bfd50-Paper.pdf)
> * Li et al. (2022): Update to cite the [ICASSP 2023 paper](https://ieeexplore.ieee.org/document/10094626)
> * Pascanu et al. (2014): Updated to cite the [ICLR proceedings version](https://openreview.net/forum?id=bSaT4mmQt84Lx)
>
> Moreover, we found a [JMLR paper](http://jmlr.org/papers/v26/24-2050.html) corresponding to the following two preprints.
>
> * Wang et al. (2023)
> * Ma et al. (2024)
>
> We have carefully proofread the entire bibliography to ensure all peer-reviewed publications are cited in their final published form rather than as arXiv preprints.
>
> ### Further Recommendations
>
> #### Citation of Yarotsky (2017)
>
> We thank the reviewer for this valuable suggestion. We have added the citation to Yarotsky (2017) to the introduction. This paper indeed provides an important complementary perspective by analyzing ReLU networks in terms of approximation error rather than the number of linear regions, and we agree it deserves mention alongside Montufar et al. (2014). Additionally, in response to comments from reviewer cVFM, we also refer to the study on the universal approximation bounds for three-layer neural networks by Barron (1993).
>
> The revised description in the introduction is as follows:
>
> > Various metrics can be considered for evaluating expressivity. For shallow NNs with three layers, Barron (1993) showed that, under certain conditions, any function can be universally approximated. On this basis, it was long believed that a depth of three layers is sufficient for NNs. Subsequently, as the superior performance of deep NNs was empirically demonstrated, theoretical researchers became interested in the advantages of increasing network depth. One of the early studies (Montúfar et al., 2014) on this topic evaluated the number of linear regions representable by NNs. This work showed that the maximum number of linear regions representable by deep NNs with ReLU activations grows polynomially in the width and exponentially in the depth of the network. Following this, various studies have evaluated different quantities related to linear regions (Pascanu et al., 2014; Serra et al., 2018; Hanin & Rolnick, 2019; Esaki et al., 2020). While these studies do not directly assess approximation accuracy of functions, it is intuitively clear that functions with an insufficient number of linear regions cannot approximate complex functions well. For instance, a function with only one linear region can not approximate a smooth curve well. Subsequently, following the approach of Barron, the approximation accuracy of functions within some classes is evaluated. For example, Yarotsky (2017) demonstrated that increasing the depth of a NN is more efficient than increasing its width for approximating functions in Sobolev spaces. Although Yarotsky (2017) did not directly used the results by Montúfar et al. (2014), his proof relies on a similar sawtooth (tent map) construction used by Montúfar et al. (2014) to count linear regions.
>
> > Thus, in the literature on the expressivity of deep NNs, the number of linear regions was historically evaluated first, then, approximation accuracy of functions was evaluated. In light of this background, we evaluate the number of linear regions of ternary NNs in this study. Our results may also provide some insights into evaluating the approximation accuracy of functions by ternary NNs, and it remains an important direction for future work.
>
> #### Clarification of Figure Labels
>
> We appreciate the reviewer pointing out the potential confusion regarding edge labels in our figures. We have revised the figure captions to provide clearer explanations of the meaning of the numbers. Furthermore, we used triple lines to represent an edge in the abbreviated notation, clarifying that it corresponds to multiple edges in the original graph. The following description is included in the revised caption:
>
> > Note that the number near by the triple lines between the square node and the round node represents not an edge weight but the sum of the original edge weights
>
> We believe these revisions will make the figures more immediately accessible to readers.

---

> > ### Author Response · Authors · 2026-03-12
> >
> > ### Additional Revision
> >
> > Please keep in mind that we added the following Remark 2 about the ternary NN with ReLU in all the layers, adressing the comments from reviewer cVFM.
> >
> > > Remark 2: The identity function can be represented by two ReLU functions as follows: $x = \mathrm{ReLU}(x) - \mathrm{ReLU}(-x)$. Using this equation, Theorem 1 can be extended to ternary regression NNs with ReLU in all the layers. For example, a finite integer weight edge as shown in Fig. 6 (a), which is the same as in Fig 3 (a), can be represented by a ternary ReLU Regression NN as shown in Fig. 6 (b). By replacing all the square nodes in Fig. 5 with this method, we can achieve a bound similar to Theorem 1 using a ternary ReLU Regression NN with double the width of the ternary NN with alternating ReLU and identity layers.
> >
> > We also inserted the following sentense to Abstruct and Conclusion.
> >
> > > When using ReLU in all the layers, a similar bound is obtained by further doubling the width.
> >
> > We hope these revisions adequately address the reviewer's concerns. Once again, we thank the reviewer for their careful reading and helpful suggestions that have improved the quality of our manuscript.

---

### Review · Reviewer_FGbB · 2026-01-30

**Summary Of Contributions:**

The authors extend a theory about the expressiveness of ReLU regression networks to the case when parameters are quantized to ternary precision (-1, 0, or 1). The proposed approach is simple: first the authors review a proof shown in previous work for continuous parameters, and then demonstrate that any continuous-parameter ReLU regression network can have its input-output relationship replicated exactly by constructing a ternary network in a specific way that involves increasing width and depth. The bound derived for the continuous-parameters can thus apply to the ternary network, yielding a new scaling relationship since the ternary network requires different numbers of parameters and layers to implement the function.

**Audience:**

Yes

**Audience Explanation:**

This work is useful in providing an intuition for how and why quantized neural networks work, what their limitations are, and what we can expect of their capabilities. This is important because compressing models via quantization is a relevant research area and is ubiquitous among practitioners, both of who comprise TMLR's audience.

**Claims And Evidence:**

Yes

**Claims Explanation:**

I believe the derivation and construction of the examples to be correct.

**Requested Changes:**

While the paper provides a tight, intuitive theory, it would be interesting to see it applied to a practical example. For example, one could investigate how well a real-world ternary network satisfies the expressivity bound derived here and, by working backwards from the authors' ternary-network construction approach, infer what a continuous-parameter version of the network would look like. These are just some of the applications of this theory. Nevertheless, while it would make the work's results more interesting, I do not think that additional experimental work is necessary for my recommendation to accept the paper.

---

> ### Author Response · Authors · 2026-03-12
>
> We sincerely thank the reviewer for the careful reading of our manuscript and for the constructive feedback. We are pleased that the reviewer found the derivation and construction to be correct, and that the work is considered useful for understanding the expressiveness of quantized neural networks.
>
> ### Suggested Experiments
>
> Regarding the suggested additional experiments, clarifying the number of linear regions in actual neural networks is also included in our motivation for this research, and we agree that such empirical investigations would be an interesting direction and could further illustrate the practical relevance of our theory. However, counting the number of linear regions in actual neural networks is computationally difficult. Therefore, we decided to abandon numerical experiments this time and instead focus on theoretically examining lower bounds.
>
> We respectfully request that the paper be considered for acceptance as a purely theoretical contribution at this stage, without additional experimental work. We also note that the reviewer has kindly indicated that additional experimental work is not considered necessary for a recommendation to accept, which we greatly appreciate. We hope the reviewer and the TMLR audience will find value in the theoretical contribution as presented, and we view the suggested experiments as a promising avenue for future work. Therefore, we added the following sentense in the section of conclusion and future work.
>
> > Applying the expressivity bound derived in this paper to a practical example is also important, e.g., investigating how well a real-world ternary network satisfies the bound.
>
> ### Additional Revision
>
> Please keep in mind that we added the following Remark 2 about the ternary NN with ReLU in all the layers, adressing the comments from reviewer cVFM.
>
> > Remark 2: The identity function can be represented by two ReLU functions as follows: $x = \mathrm{ReLU}(x) - \mathrm{ReLU}(-x)$. Using this equation, Theorem 1 can be extended to ternary regression NNs with ReLU in all the layers. For example, a finite integer weight edge as shown in Fig. 6 (a), which is the same as in Fig 3 (a), can be represented by a ternary ReLU Regression NN as shown in Fig. 6 (b). By replacing all the square nodes in Fig. 5 with this method, we can achieve a bound similar to Theorem 1 using a ternary ReLU Regression NN with double the width of the ternary NN with alternating ReLU and identity layers.
>
> We also inserted the following sentense to Abstruct and Conclusion.
>
> > When using ReLU in all the layers, a similar bound is obtained by further doubling the width.
>
> We thank the reviewer once again for the thoughtful and encouraging review.

---

### Review · Reviewer_cVFM · 2026-02-28

**Summary Of Contributions:**

This paper contributes a theoretical result, which may help us better understand the representational capacities of low-precision neural networks. In particular, this paper studies deep ternary neural networks with ReLU activations, and analyzes the number of linear regions (in the fashion of Montufar et al. 2014) that can be generated by ternary weights instead of real-valued weights. The key proof idea is to construct a ternary two-layer ReLU(+Identity) network which can express a layer with bounded integer weights; that is, one can simply double the depth to approximate bounded-integer nets with ternary nets. This result directly translates the Montufar et al. result to handle the ternary weights.

**Audience:**

Yes

**Audience Explanation:**

Theoretical understandings on what low-precision networks can express is of much interest. There has been several works on the direction function approximation side, but I have not seen much on the side of the number of linear regions.

**Broader Impact Concerns:**

This paper does not need a broader impact statement, as this work explores the mathematical nature of neural-network-configured functions.

**Claims And Evidence:**

Yes

**Claims Explanation:**

The key theoretical ideas have been presented in the main sections very clearly.

**Requested Changes:**

- As a minor suggestion: as this paper is technically simple, it will be better if authors put more effort in describing the general landscape of the current theoretical understandings on the representational capacity of low-precision neural networks. In particular, please try to make connections with Barron-style universal representation works---what are the key advantages of studying the number of linear regions, and how this can helps understanding the approximation capacities.
- In addition, please make it clearer what are the key assumptions of this work. In particular, the fact that one allows using "Identity functions" as the activations in the odd-numbered layers may be viewed as quite restrictive to the scope of this work, as many practical neural networks do not consider such activations. Also, whether the real-valued bias can be used is one of the key issues of theoretical works on low-precision neural networks. Please make a clear distinction between the "weights" and the "biases," and which assumption has been applied on each.

---

> ### Author Response · Authors · 2026-03-12
>
> We sincerely thank the reviewer for the thorough and insightful comments. We address each point below.
>
> ### Broader theoretical landscape and connections to Barron-style approximation theory
>
> We thank the reviewer for this suggestion. In the introduction of the revised manuscript, we first noted that there are various perspectives on the theoretical analysis of neural networks, including Barron's universal approximation bounds, and then explained the reason why this paper focuses on the number of linear regions, as follows.
>
> > However, the theoretical understanding of why these discretization methods work effectively remains insufficient. The motivation of this study is to provide a theoretical explanation for the success of ternary NNs. When theoretically evaluating the performance of NNs, various perspectives can be considered, e.g, the expressivity, i.e., the complexity of functions representable by NNs, the empirical error for training data, and the generalization error when applying the trained model to new data. In this work, we evaluate the expressivity of ternary NNs, as restricting the parameter space raises concerns about its significant impact on the class of functions that NNs can represent. It should also be noted that, since expressivity is defined as a property of the NN itself independently of data, this study does not consider learning from data.
>
> > Various metrics can be considered for evaluating expressivity. For shallow NNs with three layers, Barron (1993) showed that, under certain conditions, any function can be universally approximated. On this basis, it was long believed that a depth of three layers is sufficient for NNs. Subsequently, as the superior performance of deep NNs was empirically demonstrated, theoretical researchers became interested in the advantages of increasing network depth. One of the early studies (Montúfar et al., 2014) on this topic evaluated the number of linear regions representable by NNs. This work showed that the maximum number of linear regions representable by deep NNs with ReLU activations grows polynomially in the width and exponentially in the depth of the network. Following this, various studies have evaluated different quantities related to linear regions (Pascanu et al., 2014; Serra et al., 2018; Hanin & Rolnick, 2019; Esaki et al., 2020). While these studies do not directly assess approximation accuracy of functions, it is intuitively clear that functions with an insufficient number of linear regions cannot approximate complex functions well. For instance, a function with only one linear region can not approximate a smooth curve well. Subsequently, following the approach of Barron, the approximation accuracy of functions within some classes is evaluated. For example, Yarotsky (2017) demonstrated that increasing the depth of a NN is more efficient than increasing its width for approximating functions in Sobolev spaces. Although Yarotsky (2017) did not directly used the results by Montúfar et al. (2014), his proof relies on a similar sawtooth (tent map) construction used by Montúfar et al. (2014) to count linear regions.
>
> > Thus, in the literature on the expressivity of deep NNs, the number of linear regions was historically evaluated first, then, approximation accuracy of functions was evaluated. In light of this background, we evaluate the number of linear regions of ternary NNs in this study. Our results may also provide some insights into evaluating the approximation accuracy of functions by ternary NNs, and it remains an important direction for future work. The main limitations of this study are as follows. While models such as BitNet b1.58 quantize not only the weights but also the outputs of activation functions, this aspect is outside the scope of this study.
>
> While our results do not directly contribute to the evaluation of approximation accuracy of functions, we recognize that it is an important direction for future work. Therefore, we add the following description in the section of concusion and future work.
>
> > Moreover, the evaluation of approximation accuracy of functions by ternary NNs remains an important direction for future work.

---

> > ### Author Response · Authors · 2026-03-12
> >
> > ### Clarification of the assumption of the identity activation functions
> >
> > We thank the reviewer for raising this important point. We acknowledge that the use of identity activations in odd-numbered layers may appear to depart from standard practical architectures. However, we found that we can extend our results to NNs with ReLU activation functions in all the layers, because the identity function can be represented by two ReLU functions as follows: $x = \mathrm{ReLU}(x) - \mathrm{ReLU}(-x)$. Therefore, we added the following Remark 2 and Figure 6 at the end of Section 4.
> >
> > > Remark 2: The identity function can be represented by two ReLU functions as follows: $x = \mathrm{ReLU}(x) - \mathrm{ReLU}(-x)$. Using this equation, Theorem 1 can be extended to ternary regression NNs with ReLU in all the layers. For example, a finite integer weight edge as shown in Fig. 6 (a), which is the same as in Fig 3 (a), can be represented by a ternary ReLU Regression NN as shown in Fig. 6 (b). By replacing all the square nodes in Fig. 5 with this method, we can achieve a bound similar to Theorem 1 using a ternary ReLU Regression NN with double the width of the ternary NN with alternating ReLU and identity layers.
> >
> > We also inserted the following sentense to Abstruct and Conclusion.
> >
> > > When using ReLU in all the layers, a similar bound is obtained by further doubling the width.
> >
> > Moreover, we refined the usage of the term "ReLU Regression NN". In the manuscript before revision, the term "ReLU Regression NN" was used to refer to regression NNs whose activation functions are ReLU or the identity function. (We believe this caused the confusion.) In the revised manuscript, the term "ReLU Regression NN" is now used exclusively to refer to regression NNs in which all activation functions are ReLU. NNs that employ the identity function in odd-numbered layers are now referred to as "NNs with alternating ReLU and identity layers."
> >
> > ### Distinction between weights and biases
> >
> > We thank the reviewer for highlighting this issue. In the original manuscript, the distinction between weights and biases was certainly ambiguous. We believe it was caused by the following reasons.
> >
> > * The linear function of $\boldsymbol{x}$ was initially expressed as $\boldsymbol{W} \boldsymbol{x} + \boldsymbol{b}$.
> > * In Fig. 3, edge weights were represented by $w$.
> > * The words "weights" and "coefficients" were used interchangeably
> >
> > In this study, we assume that the constant term at each layer is always 1, and that the bias term is obtained by multiplying it by the edge weight. Therefore, there is no distinction between the coefficients and the intercept (bias) in the linear function, as both are expressed by edge weights. We have thus made the following revisions.
> >
> > * The linear function of $\boldsymbol{x}$ is expressed as $\boldsymbol{W} \tilde{\boldsymbol{x}}$, where $\tilde{\boldsymbol{x}} = [x_1, x_2, \dots , x_n, 1]^\top$.
> > * The word "coefficients" is unified to "weights".
> >
> > Moreover, we added the following description for Figure 1 in page 3.
> >
> > > NNs can be represented as graphs as shown in Fig. 1. Each small black dot represents the constant term 1. In this paper, we do not distinguish the coefficients for input variables and those for constant terms, i.e., biases. Both are represented by $W_{i,j}^{(l)}$ and called *weights*. In other words, if NNs are ternary, biases are also restricted to $\{ 1, 0, -1 \}$ in our setting.
> >
> > We thank the reviewer once again for the careful reading and constructive suggestions, which we believe will meaningfully improve the clarity and positioning of our work.

---

### Comment · Action_Editor_KLrz · 2026-03-10
**Discussion**

Hi everyone,

Authors --- you haven't yet responded to the reviews, and the official discussion period is about to come to an end. Could you please respond to the reviews?

Reviewers --- please allow the authors some extra time before you submit your final recommendations

---

> ### Author Response · Authors · 2026-03-12
>
> Dear Editor and Reviewers,
>
> Thank you for your patience and for the kind reminder. We sincerely apologize for the delay in our response, and we are sorry for any inconvenience this may have caused.
>
> Please find our responses to the reviews attached. We have also uploaded a revised version of the manuscript. If the system allows, we would also be happy to continue addressing any additional questions or concerns from the reviewers at any time — please do not hesitate to follow up.
>
> Thank you again for your understanding and flexibility.
>
> Best regards,
> The Authors

---

### Decision · Action_Editor_KLrz · 2026-04-08

**Recommendation:** Accept as is

**Audience:**

Yes

**Audience Explanation:**

Understanding the expressiveness of quantised networks (and in particular ternary-valued weights) is of interest to the community, and is an important research direction. All reviewers agree.

**Claims And Evidence:**

Yes

**Claims Explanation:**

All reviewers agree. Some minor and critical but easy to fix things were raised in the initial reviews, and the reviewers found the authors responses to be satisfactory.

---

> ### Author Response · Authors · 2026-04-15
>
> Dear Action Editor and Reviewers,
>
> Thank you very much for your time and thoughtful engagement throughout the review process.
> In particular, we would like to express our sincere gratitude for the fruitful discussion, through which we were able to add results for the case where all layers use ReLU activations. We believe this substantially strengthened the paper.
> We hope our work contributes to the community's understanding of the expressiveness of quantized networks.
>
> Best regards,
> The Authors